# Socio-Environmental Impacts of the Avocado Boom in the Meseta Purépecha, Michoacán, Mexico

**Alfonso De la Vega-Rivera ***  **and Leticia Merino-Pérez**

Instituto de Investigaciones Sociales, Universidad Nacional Autónoma de México,
Ciudad de México 04510, Mexico; merino@sociales.unam.mx
* Correspondence: poncho.delavega@gmail.com; Tel.: +52-1-5526853858

**Abstract:** The rapid expansion of avocado orchards in the Meseta Purépecha, in the state of Michoacán in central Mexico, has mostly been driven by the increasing demand of North American consumers in the context of the North American Free Trade Agreement (NAFTA). While the cultivation of avocado has produced regional economic growth, economic profits are strongly concentrated, notably in the hands of American agribusiness, while its regional and local environmental impacts increasingly affect indigenous and rural communities, the most vulnerable groups in that region. Our work describes the diverse impacts of avocado industrial production on local livelihoods, communal rights, and public health and also reviews the policies and incentives that have favored avocado expansion in the Purépecha region. We compared the land-use change and socio-ecological deterioration associated with avocado expansion in the majority of the Meseta with the indigenous community of San Juan Nuevo Parangaricutiro also in that region, where strong community institutions have enabled San Juan to limit avocado expansion and maintain the communal forests.

**Keywords:** land-use change; deforestation; avocado expansion; avocado production chain; social impacts; environmental impacts

## 1. Introduction

During the last five decades, avocado cultivation has grown rapidly in the Meseta Purépecha in Michoacán state, located in Central México. From 1990–2016, the area devoted to avocado cultivation in Michoacán nearly tripled, growing from 58,798 to 148,423 hectares. Over the last years, avocado is also grown in other Mexican states and other Latin American countries, but 72% of the land covered with avocado orchards in Mexico is still to be found in Michoacán [1], while Mexico covers 34% of the global demand for avocado [2].

The strong expansion of avocado production, mostly export-oriented, produced important profits, becoming the pillar of the regional economy. However, the profits of this boom are very unequally distributed, while its very significant social and environmental costs deeply affect local communities. Avocado expansion has created land dispossession in various indigenous and peasant communities, where poverty and extreme poverty remain very high, while food insecurity and health problems are common among agricultural workers, who are often community members. Avocado expansion has also exacerbated violence in an already-violent region, as "narcos" and other criminal groups, already present in the region, found, in avocado production, an ideal chance for money laundering, seeking progressively to gain control of the profitable avocado business.

The expansion of industrial agriculture during the last decades, has been an important driver for deforestation in tropical and temperate forests in the Americas, Africa, and Asia [3]. Deforestation aggravates the loss of biodiversity, destroying the livelihoods of nearby communities [4].

Michoacán is among the top five states with the highest biodiversity in México; the region is considered to be a Key Biological Area (KBA) defined as vital to the preservation

of threatened species [5]. As avocado cultivation in Michoacán basically comprises a monoculture of the Hass avocado variety, it requires a high use of pesticides and other agrochemicals that have polluted the soils and waters. Avocado cultivation has also led to the depletion of water sources in a region previously rich in water resources [6].

Different studies have assessed land-use change in Michoacán, identifying avocado expansion as the key driver of deforestation [7,8]. Others have described how avocado jumped from local to global markets [9,10], its nutritional properties and consumption patterns [11,12], and the orchard management and the pollution associated with this cropping [13]. This paper seeks to contribute to the analysis of the social conditions under which avocado production takes place and its impacts on the lives of the people of the Meseta Purépecha, an analysis largely absent in the academic literature to date. Our work focuses in the changes in land tenure and property rights, the vulnerability of the avocado production in the region, and the growth of violence related with avocado expansion. Based on a local case, we also reflect on the factors that may enable communities to minimize land-use change and deterioration.

The main questions guiding this work are:

What may have been the main drivers of the "avocado boom"? What are the most relevant social and environmental impacts of the avocado boom in the Meseta Purépecha? What are the main vulnerabilities of the avocado productive chain? What are the local governance practices that have allowed the community of San Juan to produce and export avocado, minimizing the processes of land-use change and environmental deterioration?

## 2. The Region, Methods, and Sources

### 2.1. The Meseta Purépecha

The Meseta Purépecha region includes 11 municipalities: Charapán; Cherán; Los Reyes; Nahuatzen; San Juan Nuevo Parangaricutiro; Paracho; Peribán; Tancítaro; Tingambato; Uruapan; and Ziracuaretiro. In 2020, 660,651 people lived in these municipalities. Excluding the population of the city of Uruapan, the main urban regional center, 60% of the individuals living in the Meseta recognized themselves as indigenous (Purépecha) people [14,15].

The main forest ecosystems in the Meseta Purépecha are pine forests (410,170 has), oak forests (309,787 has), and mixed pine–oak forests (822,249 has) [16,17]. Pine and mixed forests are located at an altitudinal range between 1500 and 3000 m above sea level (masl) [16]. To date, these are the forest lands that have mostly been displaced by avocado plantations [8,18].

In recent years, the cultivation of avocado has brought about deep changes in the traditional agricultural and peasant–communal culture of the Purépecha people. The Meseta Purépecha has an extension of 405,300 hectares (Figure 1). Traditionally the majority of the lands were occupied by forests and "milpas", that is, cornfields, with the presence of beans and squash devoted to familial consumption and to local markets [19]. Coffee, produced in the most humid and low lands, was the main cash crop in the traditional regional economy.

Field work for this research was carried in the municipalities of Uruapan and San Juan Nuevo Parangaricutiro, both with large extensions of avocado orchards. The municipality of San Juan Nuevo has 18,834 has and a population of 20,981 people, and Uruapan has with an extension of 315,350 has and a population of 356,786 inhabitants.

These data show that, despite decades of avocado production and the important wealth created, poverty and extreme poverty prevail in this region with an important indigenous presence and a very young population. San Juan Nuevo is a municipality with a mainly indigenous population, much of them very young. Despite decades of avocado production, 68% of the population is poor, and more than 11% is extremely poor. Uruapan has an annual Gross Domestic Product (GDP) much higher than that of the state of Michoacán and higher than that of Mexico (Table 1), but, to date, the majority of the population is poor, and nearly 10% is extremely poor. In the Meseta, more than 14% of persons are is poor, and 63% is poor.

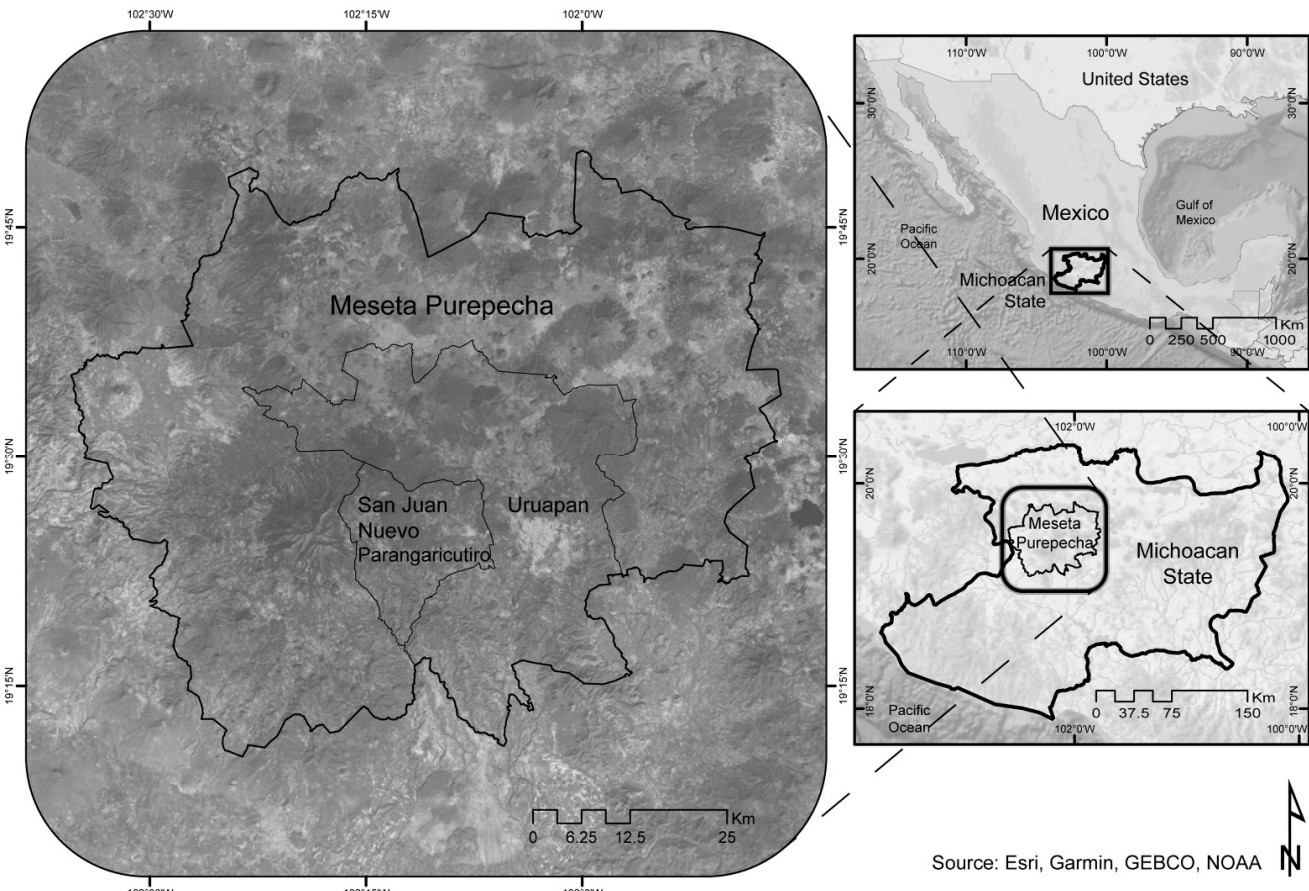

**Figure 1.** Geographical location of the Meseta Purépecha and the municipalities of San Juan Nuevo Parangaricutiro and Uruapan.

**Table 1.** Population data of the municipalities of Uruapan and San Juan Nuevo, la Meseta Purépecha, the State of Michoacán, and the country of Mexico. Compiled by authors based on the data from [14,15,20,21].

| | Population | Average Age | Indigenous Popula-tion | Human Development Index [1] | GDP per Capita (USD) | Population Living in Poverty % [2] | Population Living in Extreme Poverty % |
|---|---|---|---|---|---|---|---|
| Nuevo Parangaricutiro | 20,981 | 26 | 67% | 0.65 | 8028 | 68% | 10.6% |
| Uruapan | 356,786 | 27 | 19% | 0.73 | 12,242 | 56.4% | 9.3% |
| Meseta Purépecha | 660,651 | 24 | 32% | ND | ND | 63.4% | 14.3% |
| Michoacán | 4,748,846 | 28 | 14% | 0.69 | 5147 | 59.11% | 9.92% |
| Mexico | 126,014,024 | 26 | 7% | 0.76 | 9271 | 47.54% | 8.37% |

[1] The Human Development Index (HDI) is a statistical composite index of life expectancy, education (literacy rate, gross enrollment ratio at different levels, and net attendance ratio), and per-capita income indicators, which are used to rank countries in four tiers of human development. [2] The data referred to population living in poverty do not include those of the population living in extreme poverty.

Data on inequality (Table 2) based on the Palma Index [22], which assesses the income of the 10% of the population with the highest income in relation to the lowest 40%, show an important income concentration in the municipality of Uruapan, the main urban center of the region, much more unequal not only than San Juan but and the state of Michoacán as a whole. Inequality in Uruapan is even higher than inequality in Mexico, a deeply unequal country, expressing the strong concentration of the gains of avocado production. In contrast inequality in San Juan Nuevo is very low, while the majority of the population is poor, as the major avocado producers in San Juan municipality live in Uruapan or outside Michoacán.

**Table 2.** Margination and inequality indexes in the municipalities of Uruapan, San Juan Nuevo, and Michoacán State. Estimated by authors based on the data from [20,23].

| | Palma Index in 2010 |
|---|---|
| Uruapan | 3.05 |
| San Juan Nuevo | 0.46 |
| Michoacán State | 2.9 |
| Mexico | 2.8 |

An important extension of the lands of the region and in the two municipalities under study is the collective holdings (agrarian communities and ejidos, Warman 1990). Many lands are covered with avocado orchards in both collective and private lands.

The majority of avocado packing facilities are found in the city of Uruapan, whose production is oriented to international and national markets. In Uruapan, there is also a business of avocado processing and even an international airport.

*2.2. Methods*

We selected the municipalities of Uruapan and San Juan Nuevo due to the existence of several conditions in common: an important presence of Purépecha people living under poverty and deprived conditions; important extensions under collective property; similar environmental conditions and high deforestation rates. The selection of the municipality of Uruapan and the field work in the city of Uruapan enabled us to contact key agents in the processing and marketing of avocado production. Our work in the community of San Juan Nuevo Parangaricutiro provides information on a case in which communal institutions are vital for more sustainable outcomes.

1.  Between November 2016 and March 2018, 33 in-depth semi-structured interviews [24] were carried out in the towns of Arandín, Milpillas, and San Juan Nuevo in San Juan's municipality, and in the town of Capácuaro, and Uruapan City in the municipality of Uruapan. The interviews were based on a semi-structured questionnaire (included in Appendix A), which was elaborated based on the methodology proposed by Kallio et al. (2016) [24] and applied to key informants, who were based on the previous knowledge of the region and on the "snow ball" sampling technique. This is specifically used for individual interviews and is a type of deterministic sampling method. In this technique the first interviews are applied to a group of key informants previously identified (these were originally eight people in our case study) asking them to recommend other potential interviewees who from their perspective are also relevant actors in the process under study, and so on, aiming to reach a relevant number of interviews until the responses become consistently repetitive [25,26]. We choose this sampling method as it allowed us to reach key informants between populations difficult to access [27], due to the prevailing mistrust among avocado producers, government officers, and community authorities due to the generalized violence, extorsions, and kidnappings in the region committed by the organized crime. The "types" of actors that we interviewed were: Twelve small- and medium-scale farmers who own and/or rent private land where they grow avocado; eight sanitary technicians, in charge of the registration and authorization of avocado cutting and shipping of export permits to the US; two municipal (government) authorities of both San Juan Nuevo and Uruapan; the president of the indigenous community of San Juan Nuevo, five agricultural workers, and four regional experts in the themes of: forestry, water, and land-use change. The number of the different actors interviewed and the size of the whole sample were defined based on the repetitiveness of the information gathered in the different interviewees [27]. These interviews provided qualitative information, critical for the understanding of the process under analysis, based on the perspectives of different stakeholders and relevant actors.

2. For the analysis and grouping of the ages of the orchards and land tenure, we conducted an overlay analysis with the software ArcMap ver. 10.3 using the data of the Study of Assessment of Ecological Impacts of Avocado Cultivation, at the Regional and Plot Level, for the years 1995, 2005, and 2011 by Burgos et al., 2011a, 2012 [7,8] and the data on land tenure provided by the Registro Agrario Nacional [28].

3. This work is also based on the analysis of different documental sources: the 2000, 2005, 2010, and 2020 Population and the Agricultural Censuses of the National Institute of Statistics, Geography, and Information Technology (INEGI) [14,15,23]; the Human Development Index drafted by the United Nations Development Program [29], the National System of Information on Market Integration of the Ministry of Economics of Mexico [30]; the statistical database of the Food and Agriculture Organization (FAOSTAT) [8], and the Agri-Food and Fisheries Information Service of the Ministry of Agriculture, Livestock, Rural Development, Fishing, and Food of the United States of America [31]. This diverse information enabled a comprehensive characterization of the social and economic context of the process under study.

## 3. Results

### 3.1. The Expansion of Avocado in Michoacán

In addition to the growing presence of Mexican avocado in the international market, particularly in the US, where per capita avocado annual consumption more than tripled from 1.1 to 3.6 kg between 2001 and 2017 [32], Mexico is the country with the highest yearly avocado consumption per capita of 10.2 kg per year [30]. The recent history of avocado cultivation dates to the 1950s, with the introduction of the Hass variety in Michoacán, with the highest market value due its high productivity throughout the year and the thick consistency of its peel that facilitates its transportation and storage. By the end of the 1950s, the area devoted to avocado cultivation in Michoacán reached 15,000 hectares, located mainly on private lands in the municipality of Uruapan.

The climatic conditions and volcanic soils of the Meseta Purépecha produce a high quality and yield of avocado cropping [33]. In 1961, the Mexican Institute for Coffee (IN-MECAFE) promoted a program of crop diversification in the Meseta, mixing avocado trees with coffee plants, aiming to halt the overproduction of coffee and to protect coffee prices. Later in the 1970s, the government of the Uruapan City promoted avocado plantations as part of a program of soil-erosion control in lands originally covered with pine–oak forests, cleared in previous years [34].

Figure 2 depicts the constant expansion of lands covered with avocado plantations from 1980–2019 in Michoacán, driven by a pronounced increase in the value of the avocado.

Between 1990 and 2016, the area devoted to avocado in Michoacán grew from 58,798 to 148,423 hectares. By 2018, 72% of all the lands covered with avocado orchards in Mexico is found in Michoacán [35]. In 2018, Mexican avocado production was 33% of the 5,689,985 tons produced worldwide [8].

The commercial opening of the US market to Mexican avocado in 1997 [36] took place after the implementation of NAFTA in 1994. The US demand soon became the main driver of the rapid expansion of avocado cropping in Michoacán. Since 1997, the expansion of avocado orchards has been constant. From 1997 to 2018, the area occupied by avocado grew by 217%, moving from 76,464 has to 166,603 has [37]. From July 2019 to June 2020, Michoacán exported 962,000 tons of this fruit to the US, equivalent to 58% of the Mexican avocado production [38].

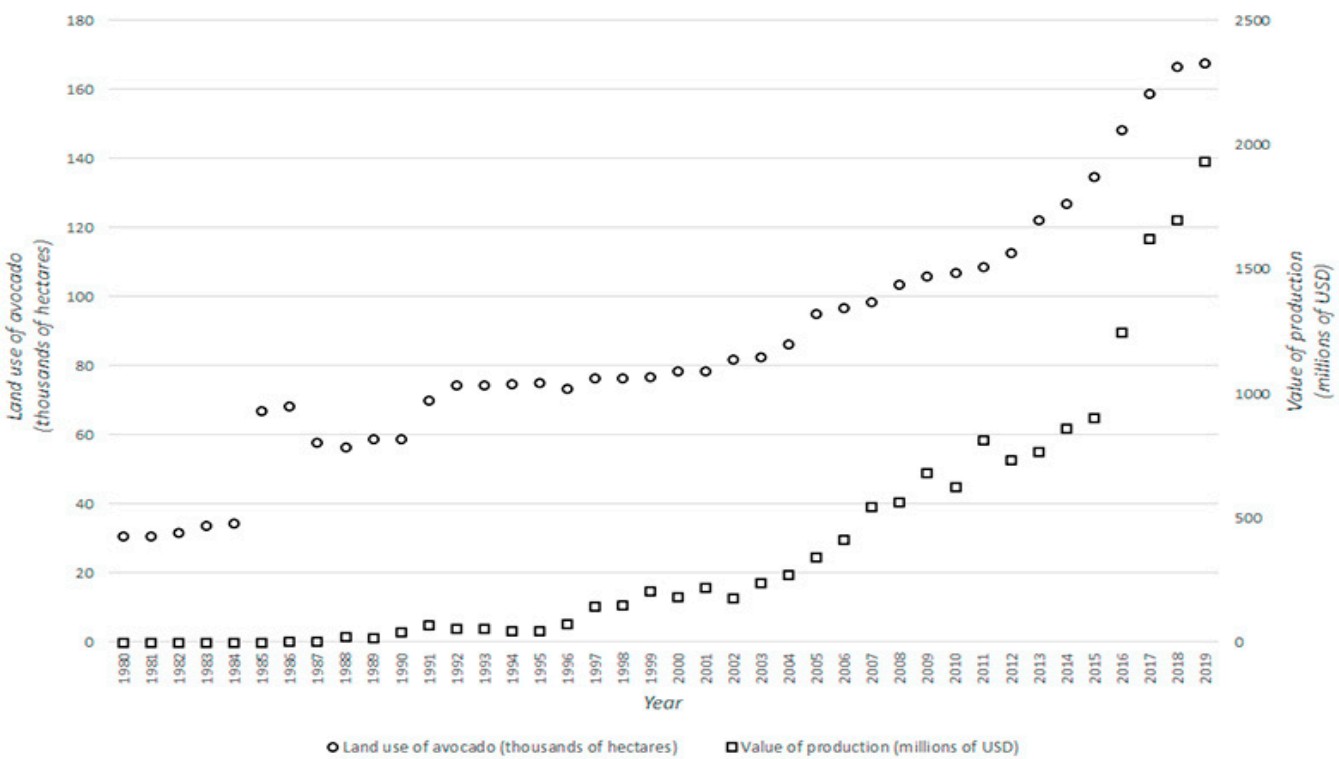

**Figure 2.** Land use and the value of production of avocado in Michoacán State from 1980 to 2019.

Avocado farmers in the US initially opposed the entry of Mexican avocados but ended up benefiting from it, as per capita consumption and the price of avocado have constantly increased in North America. According to producer Ricardo Olivo "American farmers benefited from the entry of Mexican avocado because the Mexican avocado exporters association has invested a lot of money to position the product in the taste of consumers in the US". (Ricardo Olivo, personal communication, 26 October 2017). Avocado became an important ingredient in the US culinary culture, reaching an estimated consumption of 100,000 tons during the 2017 edition of the Super Bowl [39]. In less than 20 years, Mexican exports of avocado to the US increased more than 13,000-fold, from 6032 tons in 1997 to 790,920 tons in 2016–2017 [39].

Since Mexico entered NAFTA, agricultural policies shifted in favor of export crops controlled by agribusiness. Mexico's government abandoned policies of support of small farmers traditionally oriented to the production of staple crops for the national market. Different legal reforms enabled the privatization of ejido lands and water resources in favor of large farmers and corporations [40,41].

The key incentive for land-use change for landowners is the high-opportunity cost of forest conservation with respect to avocado production. Comparison of the gains of forest production, mostly commercial logging, and avocado cultivation shows deep differences. The highest forest productivity in the region, achieved in San Juan Nuevo, ranges from 19–71 m$^3$/ha of timber, while the average price of one cubic meter of pine round wood in 2018 was 61 USD [42], with profits from forest production between 1160 and 4300 USD per hectare; the average seasonal yield of an avocado orchard is 7 tons/ha for orchards of fewer than 10 years and 13.5 ton/ha for older orchards [37]. Export prices per ton ranged from 2300 to 2700 USD in March 2021 [43], producing profits ranging from 16,100 to 18,900 USD for orchards with less than 10 years of age, ranging from 31,050 to 36,450 USD for older ones. In addition to this pronounced economic difference, avocado orchards can be harvested as early as 4 years after they are established, while forest-cutting cycles in the region have a duration of 10 years in forests under authorized forest management.

*3.2. Impacts of Public Policies on Avocado Expansion*

Diverse policies have strongly favored avocado expansion, starting with the 1992 Amendments of Article 27 of the Mexican Constitution, which legalized the parcel and private titling of ejido lands, together with the 1992 Forest Law, which reduced the governmental inspection of timber production, contributing to the increase in deforestation in favor of avocado orchards.

A high use of fertilizers, herbicides, and insecticides was not only allowed but promoted in order to increase productivity and pest control, an important problem in monocultures. Agrochemicals are abundantly used throughout the whole production process.

A vast forest land-use change, defined as an environmental crime in Article 28 of the General Law of Ecological Equilibrium and Environmental Protection, has occurred with total impunity. For more than 25 years, despite the disappearance of at more than a third of the forest cover of the Meseta, not one single legal authorization for forest clearing was issued in Michoacán, a clear indicator of the illegal status of the majority of the avocado orchards, according to the President of the San Juan Nuevo community "Most of the 50% of the forest that surrounded the community of San Juan Nuevo no longer exist".

Credits and tax benefits for avocado producers were widely available for avocado producers, contrasting the scarce support and overregulation faced by forest producers. In addition, while forest subsidies are granted to communities, recipients of governmental support to avocado production are individuals. This subsides are captured by large- and medium-sized avocado producers.

Another meaningful difference between forestry and avocado cultivation is the important dissimilarity of transaction costs. In total, 60.3% of forests in Mexico is collectively owned, by ejidos and communities [44] and is legally defined as commons. Forest management and forest production in those forests are, by law, a communal/ejido activity, requiring collective organization and providing collective profits. Avocado cultivation is a private activity, one that is privately financed and organized. Avocado growers comprise relatively few individuals in the region, as this activity demands high investments. On the other hand, while forestry is a strongly regulated activity in which producers must finance periodical forest inventories and management plans, required in order to obtain yearly logging permits, granted by the environmental federal authorities; avocado cultivation is, in fact, nonregulated, despite its intense and damaging use of natural resources.

*3.3. Main Social Impacts: Concentration of Lands, Productive Capacities, and Profits*

Avocado cultivation is not available to all farmers, but only to those few with enough economic capacity to finance the establishment and care of the orchards for at least 4 years prior to the first harvest (Table 3). Only after 10 years do orchards become completely productive.

This high initial financial demand has led to a high concentration of avocado production in the hands of large farmers and even criminal groups. Up to now, they are the regional groups that have mostly benefited from the avocado boom. According to interviewees, many people who have opted for the change of land use to avocado orchards on their lands have requested loans from people linked to organized crime, which in many cases have taken over the land or have forced them to pay constant extortions.

After the first years of the avocado expansion, when private lands able to be converted in avocado orchards became scarce, growers moved to ejido and communal lands, previously used for domestic agriculture. Michoacán has an area of 5,986,400 hectares, of which 47% (2,786,699 ha) are collective holdings (Table 4). While only 19% of mature avocado orchards (more than 16 years of age) were established on communal/ejido property by 1994, this share has increased to 43% in 2016 of the total lands used by the most recently established orchards, (less than 6 years). Figure 3 shows the share of private vs. collective ownership of avocado orchards of different ages, in the Meseta Purépecha, and in the municipalities of Uruapan and San Juan Nuevo.

**Table 3.** Estimated average production costs and profits per hectare of avocado plantations in 2018 in the Meseta Purépecha. Based on data from: [36,43,45].

| | Cost and Profits from the Orchards in the Year 1 (USD)/ha | Cost and Profits from the Orchards in Years 2–4 (USD)/ha | Cost and Profits from the Orchards in Years 4–10 (USD)/ha | Cost and Profits from the Orchards after 10 Years and more (USD)/ha |
|---|---|---|---|---|
| Tree planting | 44.5 | 0 | 0 | 0 |
| Fertilizers | 2800 | 2800 | 2800 | 2800 |
| Maintenance and care of the plantation | 800 | 800 | 800 | 800 |
| Irrigation | 400 | 400 | 400 | 400 |
| Control of pests and weeds | 1180 | 1180 | 1180 | 1180 |
| Costs of participation in the export program, agricultural insurance, and administrative costs | 900 | 850 | 850 | 850 |
| Total | 6125 | 6030 | 6030 | 6030 |
| Sales | 0 | 0 | 7600–11,800 | 14,700–22,700 |
| Balance | −6125 | −6030 | 1570–5770 | 8690–16,670 |

**Table 4.** Land tenure and surface in avocado production in the municipalities of Uruapan, San Juan Nuevo, and the Meseta Purépecha. Estimated by authors based on data from: [8,10,21,28,35].

| | Total Extension (Hectares) | Communal-Ejido Lands (%) | Extension of Lands Covered by Avocado Orchards (has) |
|---|---|---|---|
| Uruapan | 101,500 | 39.6 | 16,200 |
| San Juan Nuevo | 23,500 | 55 | 7520 |
| Meseta Purépecha | 413,716 | 28.5 | 76,889 |

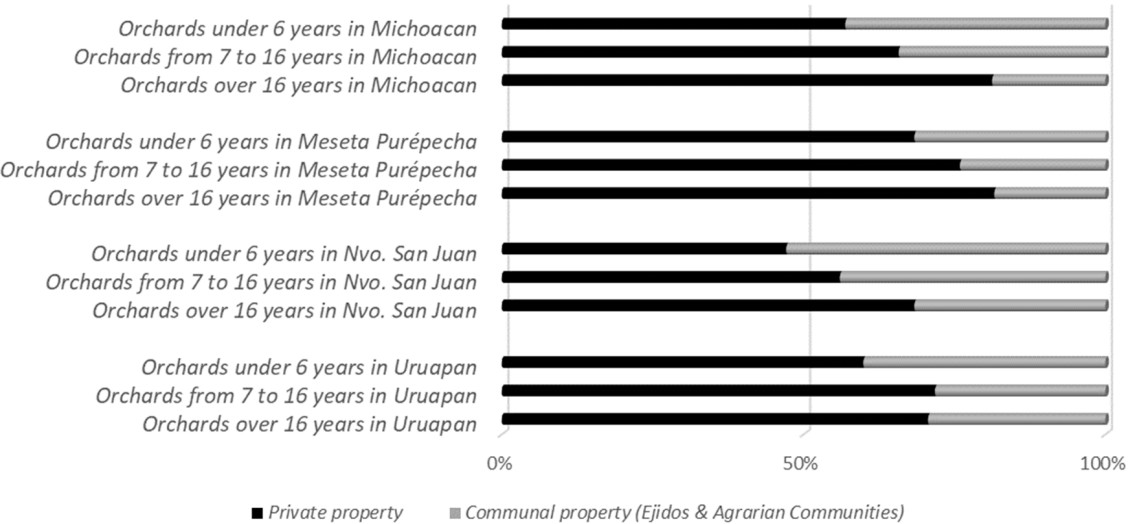

**Figure 3.** Share of avocado orchards of different ages in private and communal lands in Michoacán, the Meseta Purépecha, Uruapan, and San Juan Nuevo municipalities. Estimated by authors based on [8,10,21,28,34].

Together with the advance of the avocado plantations on community lands, in recent years, new avocado orchards tend to be established in previously forested lands, making avocado expansion one of the main drivers of deforestation in Mexico. The substitution of communal/ejido forests by privately managed avocado plantations weakens community's governance and territorial management.

Recent parcellation of the commons and land-use change are particularly pronounced in communities of the municipality of San Juan Nuevo, where already 53% of the newly established orchards occupy communal/ejido lands and 31% of the mature orchards of 10 years and more are found in communal lands. In addition, in the Municipality of Uruapan and in Michoacán as a whole, newly established avocado plantations can be frequently found in communal lands.

Legally, the access to lands through sale or rent are more difficult to obtain in indigenous communities' lands, whose privatization faces more legal requirements than ejidos' lands, as the sale of communal lands to outsiders is considered illegal. The disappearance of indigenous communities requires the decision of the majority of their members first to become ejidos and later the majority's acceptance to the parcellation and privatization of the land.

For avocado cultivation, even if lease of communal lands is also prohibited, commoners who rent lands maintain formal land ownership but lose the de facto rights of control and use [46]. Lands are usually leased for periods of 10–15 years, with the option of renewal. Land owners receive payments in advance, equivalent to the rent of the land for 3–5 years. In 2017, these payments were between 1000 and 5000 USD per hectare per year depending on the characteristics of the lands and their proximity to roads [37]. In the context of wide poverty in the Meseta and in the municipalities considered in this work, these payments are attractive for landowners but are only sufficient to cover the needs of their families for not longer than 1 year. Loss of land rights weakens family's food security, forcing their members to hire their workforce. As the producer Ricardo Olivo said, "it is common to find indigenous people working in the avocado orchards established in the lands they formally owned" (Ricardo Olivo, personal communication, 26 October 2017).

Another important change is the growing presence of criminal groups in the avocado business. Due to its high profitability, avocado cultivation is not only an ideal mean for money laundering but an attractive activity that these groups increasingly seek to control.

### 3.4. Main Environmental Impacts: Land-Use Change, Water and Soil Pollution, and Forest Fragmentation

Unlike the concentration of gains of the avocado production, the environmental externalities it creates are suffered by the entire region, affecting mostly those that already are vulnerable.

As already mentioned, avocado orchards strongly compete with pine–oak forests of the region that provide important ecosystemic services [33,47]. The optimal altitudinal range for avocado in Mexico is located between 1800 and 2200 m above sea level (masl), the same altitudinal range where temperate forests are found. During the last two decades, this forest ecosystem has rapidly been displaced; avocado expansion has become the main driver of the loss of temperate forests in Michoacán [48,49].

Avocado cultivation is currently based on the monoculture of the Hass variety, rendering orchards highly vulnerable to pests. Orchard management largely relies on a high use of agrochemicals, creating serious problems of pollution of water and soils, leading even to public health problems [50].

Orchards consisting mainly of coetaneous plantations use very large quantities of water, exerting strong pressure on water bodies, affecting the access to water of the local population and subsistence agriculture in a previously water-rich region. Chávez-León et al. (2012) and Tapia et al. (2011) estimate that, under similar conditions of vegetation coverage and age, the level of water runoff and the interception of rain in avocado orchards and in pine–oak forests are similar. However, avocado orchards have a stronger water demand due to evotranspiration, which increases during the dry season, when the orchards require

at least 700 m$^3$ of water per hectare. The water used in the orchards is extracted from the springs, wells, and rivers of the region, where the water runoff of the streams and springs is now significantly diminished. Of the total of 26,658,186 m$^3$ of water extracted annually from wells in the Meseta registered by the National Commission of Water, 69.7% is employed in agriculture, mainly in avocado orchards [47]. Due to the water concession system established in Mexico at the time of the implementation of NAFTA, water utilized in orchards has low or no costs. This is another important subsidy for avocado production and exports at the expense of the regional ecosystems and the human right to water, granted by the Mexican Constitution (Article 6).

Together with deforestation, the avocado boom has created forest fragmentation, the loss of species of flora and fauna, many of these at risk and/or endemic to the region, erosion, and loss of soil. It has also reduced the region's capacity to contribute to the mitigation of Global Climate Change and to adapt to it.

### 3.5. Vulnerability of the Avocado Production System

The avocado production system in the Meseta Purépecha is extremely vulnerable in ecological, social, and economic terms. This is due, first, to its strong dependence on the US market, which consumes more than 85% of the regional avocado production. In addition, the saturation of the international markets related to the increasing global production driven by continuously growing prices poses permanent risks of price falls, as has occurred with many other globally traded agricultural products, such as coffee, cotton, and sugar, just to mention a few.

In addition, farmers who export avocado must comply with a series of Mexican regulations, such as NOM-066-FITO-1995, and international regulations that force them to handle orchards in a very strict and particular manner. If irregularities in orchard management occur or if contaminated fruits with banned substances are detected, the trade with the US of all of the farmers from Michoacán can be summarily halted.

The environmental vulnerability of the system related to monocropping and the exclusive use of the Hass avocado variety have given rise to genetic homogeneity and increasing vulnerability to the risk of pest and diseases. Prior to the avocado boom, peasants in the Meseta Purépecha maintained an important agrobiodiversity in the agricultural fields with maize, beans, squash, and a high diversity of fruits, tubers, and herbs, including different varieties of avocado trees.

Since the beginning of the avocado boom, agricultural practices have drastically changed, leading to a landscape dominated by Hass avocado trees. This change entails phytosanitary implications: such as a constant presence of pests and a permanent need for a large use of agrochemicals. The use of water has become much more intense, leading to the overexploitation of water resources and basins in the region.

The increasing incidence of extreme climatic events, due to the processes of global climate change, has exerted a negative impact on avocado production: the presence of frosts delays or even inhibits tree flowering and the falling of hail, which damages the avocado flowers, reducing tree productivity. In 2016, hailstorms in large areas of the Meseta caused large harvest losses, leading to important increases in prices and to shortages of avocado in the national market.

From a social perspective, a process that produces vulnerability of the avocado system comprises the strong violence present in the Purépecha region for more than one decade, which has triggered the extortion of farmers and packers, many of whom have left Michoacán. Some of the wealthiest farmers have moved to other regions within Mexico, opening new avocado-producing areas in the states of Jalisco and Nayarit, the regions with the greatest expansion of avocado cultivation in recent years.

### 3.6. An Alternative Model: The Community of San Juan Nuevo Parangaricutiro

In the middle of "avocado country", in the municipality of San Juan Nuevo, the Purépecha community of San Juan Nuevo Parangaricutiro (SJNP) has developed a model of avocado

cultivation and land governance that is a marked contrast with the landscapes dominated by the avocado agribusiness in the rest of the region. The community of San Juan has an area of 18,138 ha [51], 10,000 has of which are covered by communal pine and oak forests and 2000 hectares are used for agriculture and grazing [52].

San Juan has 1229 communal right holders, known as "comuneros" who take part in the communal assembly, the main local authority. From the late 1970s to date, this assembly took the reins of forest administration, creating a communal forestry-production initiative [52]. Forestry is guided by San Juan's own technical team, enabling SJNP to be the first community in Mexico and in the world to obtain, in 1997, the Forest Stewardship Council certification for the sustainability management of SJNP forest management. For decades, this community has successfully managed the forest, sustained forest production, and produced high-value-added and high-quality products such as wooden floors, panels, and furniture with access to national and international markets [51,53]. Communal forestry provides all community members and their families with employment and income. A large share of community forest gains is invested in local public goods such as schools, clinics, and street pavement, contributing to local wellbeing.

Through this process, community institutions have been strengthened by means of an informed decision and rulemaking system based on the continuous functioning of the communal assembly. This is particularly true with issues related to forest management and with the administration of communal enterprises, the main drivers of the San Juan's economy. All commoners have an equal right to participate in local decision making on community matters (Juan Manuel Esquivel, personal communication, 16 October 2017).

During the first decade of the 21st century, the net yearly income of the forest enterprises ranged between 5.5 and 6 million USD, with an average profit of about 10% [54]. These gains are shared among San Juan commoners.

The lands of the SJNP are especially suitable for avocado cultivation. Thus, the Assembly decided to allow avocado cultivation on approximately 2000 hectares of lands with the traditional agricultural use at 2400 masl. Forest parcellation is prohibited by the community's rules, protecting forests from use change, as the president of the community said, "if it is a forest, it remains a forest" (Juan Manuel Esquivel, personal communication, 16 October 2017). Another key agreement, with a definite importance for forest conservation and community well-being, is the prohibition of the sale or transfer of community property rights to outsiders.

Most of avocado farmers in San Juan own parcels of less than 1 ha; the community assembly recognizes private rights to agricultural plots and houses, which formally are part of the communal lands. The absence of land concentration has prevented strong socio-economic differentiation. For more than 20 years, SJNP has also managed 220 has of collective avocado orchards, producing for export markets. The profits of this communal venture are totally reinvested in communal forestry enterprises as well as in other productive community projects, such as the production of strawberries in greenhouses, enabling the creation of more jobs and investment in forest management and protection; in the words of one San Juan's commoner: "Forest gives us life, avocado doesn't."

Despite being surrounded by a regional landscape of private avocado orchards, SJNP continues to preserve the communal forests, which provide vital environmental services for the entire region. Comuneros from SJNP are clear that the future of the community depends on the conservation and diversification of forest production in their lands. They are also aware that their success is largely due to social cohesion and to the strength of community governance institutions.

From the 1970 to the mid-1990s, the expansion of small domestic agriculture, often based on slash-and-burn practices, entertained an important weight in forest losses. As these practiced diminished in different traditionally peasant and indigenous regions, some authors proposed a Forest Transition [55–57] characterized by the afforestation of previous agricultural lands that would take place in the developing world, mirroring processes that took place decades or even a century ago in regions of the developed countries. In the

Meseta Purépecha, in Michoacán, this has not been the case, as the lands and resources of the region became strongly oriented to production devoted to export markets.

From a technocratic perspective, avocado is the model export whose expansion should be continuously promoted. In 2017, the Mexican government estimated avocado harvest grew from 2.05 million tons in 2018 to 2.61 million tons in 2024 and 3.16 million tons in 2030. According to this plan, and regardless of the environmental costs implied, the process of expansion and land-use change would continue, not only in Michoacán, but anywhere in the country with suitable conditions for this crop [40], this means the temperate forests of the country, the most abundant type of forest ecosystems in Mexico. This perspective evidently disregards the weight of the so-called externalities, environmental, and social impacts imposed to the members of indigenous communities, that have often become precarious agricultural workers in their own lands, without indigenous nor labor rights.

Among social impacts, concentration of the important wealth created by avocado production and marketing has come at the expense of losses of communal property rights, of food security, and access to natural resources basic for livelihoods for indigenous and peasant families in the Meseta Purépecha. Disparity and inequality are partially expressed in the changes of the values of the Palma Index for the municipality Uruapan. On top of this, the increased presence of organized crime in the region and in avocado production chain is a heavy cost, mainly suffered by those less able to move, as the main owners of avocado plantations have done.

In many areas of Mexico and Latin America, communal tenure and communal governance are viable schemes for environmental and social protection [58–61]. In this sense, the governance of forest commons has important public values as it sustains forest provision of key environmental services. The expansion of avocado in Michoacán has contributed to eroding communal institutions. Despite legal definition in Mexico of forest lands as commons, owned by ejidos, and indigenous communities, whose parcellation is prohibited by law, the advance of the avocado is based on de facto parcellation and privatization of forest commons in the context of a complete absence of enforcement of agrarian and environmental laws and indigenous rights. Through the years, this process has weakened the communities' territorial governance in many indigenous and non-indigenous communities in the Meseta, being stronger in communities with poor organization and social capital [59].

The commercial, environmental, and social vulnerability of the avocado system, the increased inequality and presence of crime in the region as extreme outcomes of a supposed model export openly express the unsustainability of global food production chains blind to environmental and social costs.

## 4. Conclusions

Industrial agriculture, largely oriented toward global markets, rapidly expanding at the expense of both forest areas and areas previously devoted to small domestic agriculture has become the main driver of deforestation in the tropics. The rapid growth of avocado cultivation in Central Mexico after the implementation of the North American Free Trade Agreement (NAFTA) is an iconic case of this processes.

The high dependence of the Meseta Purépecha on avocado production renders the region highly vulnerable in economic, social, and environmental terms. Those vulnerable are not only those directly involved in avocado production process, but for the whole Meseta, to the extent that today's regional economy is regarded as unviable in the absence of the avocado industry. In this context, the need for economic diversification and changes in the agroindustrial avocado production should be taken into serious consideration by regional and federal governments and by society.

Inequality in the distribution of costs and benefits of avocado production has created an increasingly conflictive regional situation. This is because, as previously noted, small farmers, agricultural workers, and local communities receive a minimal proportion of the important profits of this business but undergo the various "externalities" of pollution,

violence, disposition, loss of livelihoods, erosion of community governance and cohesion, and problems of public health.

Specific policies targeted to protect community rights, to promote communal organizations and cooperatives, and land and forest governance are important means for protecting indigenous and local communities and those more vulnerable within them. The experience of the community of SJNP shows the potential of collective action around forest commons as means to contain environmental destruction and halt the increase in inequality and the loss of cohesion, exhibiting avenues that should be supported by policies committed with the promotion of sustainability in indigenous regions such as the Meseta Purépecha.

The enforcement of the state of law through the compliance with environmental law and regulations, sanctioning forest land-use change, and controlling the use of agrotoxics and of onerous water use, together with the promotion of agroforestry and organic avocado production are important means to address the regional ecological crisis driven by the agroindustrial avocado production.

Markets, whether local or international, could play key roles. National consumers and consumer countries should be able to cover part of the costs of this change, assuming the internalization of these costly externalities through certification schemes.

**Author Contributions:** Conceptualization, A.D.l.V.-R. and L.M.-P.; methodology, A.D.l.V.-R. and L.M.-P.; validation, A.D.l.V.-R.; formal analysis, A.D.l.V.-R.; investigation, A.D.l.V.-R. and L.M.-P.; resources, A.D.l.V.-R.; data curation, A.D.l.V.-R.; writing—original draft preparation, A.D.l.V.-R. and L.M.-P.; writing—review and editing, A.D.l.V.-R. and L.M.-P.; supervision, L.M.-P. All authors have read and agreed to the published version of the manuscript.

**Funding:** This research was funded by the Consejo Nacional de Ciencia y Tecnología from the Mexican government.

**Institutional Review Board Statement:** Not Applicable.

**Informed Consent Statement:** Not Applicable.

**Data Availability Statement:** Demographic data can be downloaded from Instituto Nacional de Estadística y Geografía de México (INEGI) (https://www.inegi.org.mx/), (accessed on 24 March 2021); Catastral and ownership data can be downloaded from Registro Agrario Nacional de México (RAN) (requires previous registration) (https://phina.ran.gob.mx/), (accessed on 19 September 2018); Poverty indexes can be downloaded from Consejo Nacional de Evaluación de la Política de Desarrollo Social de México (CONEVAL) (https://www.coneval.org.mx/), (accessed on 21 May 2018); The data of areas produced and harvested and average prices of avocado can be downloaded from Secretaría de Agricultura y Desarrollo Rural de México (requires previous software installation), Available online: https://www.gob.mx/siap/documentos/siacon-ng-161430 (accessed on 12 February 2020); The data of market prices of avocado for exportation can be consulted in Asociación Agrícola Local de Productores de Productores de Aguacate en Uruapán Michoacán, México (https://aproam.com/precios/), (accessed on 3 April 2021).

**Acknowledgments:** The first author extends acknowledgment to the PhD Program in Sustainability Sciences, UNAM "Doctorado en Ciencias de la Sostenibilidad, Universidad Nacional Autónoma de México.

**Conflicts of Interest:** The authors declare no conflict of interest.

## Appendix A

Script for the semistructured interview.

I.  Production
    1. How long have you been growing avocado?
    2. What did you do before producing avocado?
    3. How did you start growing avocado?
    4. What is the area that has been sown?
    5. Do you produce any other produce in the orchard?
    6. What is the yield of your orchard?

7. How many times do you harvest in a year?
8. Has yield changed in recent years?
9. What variety of avocado do you produce?
10. Where did you get the seedlings to plant the orchard?
11. How do you manage the orchard?

Conventional
Organic (to question 16)

12. Secondary vegetation removal
13. Use of Herbicide

Which?

14. Fertilizer

Which?

15. None
16. What is the origin of your organic inputs?
17. Which ones do you use?

a.     Bordeaux broth
b.     Sulfo-calcium
c.     Bocachi
d.     Lombri-compost
e.     Humus
f.     Others

18. Does your orchard have irrigation? No (to question 22)
19. How much water do you use to irrigate?
20. Where does the water you use to irrigate come from?
21. Is water available throughout the year?
22. Do you require electricity for your production process?
23. Do you know roughly the cost of producing one ha per year?

Water consumption:
Fertilizer consumption:
Phytosanitary control (herbicides/organic inputs):
Machinery and equipment:
Other:

24. Do you have any certification?

Good practices
Organic
Export

25. What are the advantages of these schemes?
26. What are the disadvantages of these schemes?

II.    Commercialization
27. Who do you sell it to?
28. How do you sell it?
29. Do you know if it is exported?

Where?

30. Have you always sold it to the same people?
31. It belongs to an organization of producers/marketers (No to 35)
32. How long have you been with the organization?
33. What are the advantages of belonging to the organization?
34. In your experience, what is the reason for the avocado boom?

III. Property regime

35. Is the orchard yours, is it part of an ejido, is it private property, is the rent?
36. Does your orchard belong/belonged to any ejido or community?
37. Do you know what used to be produced on the land where you have your avocado orchard? (No to question 39)
38. When was the substitution made?
39. Why was the crop substituted?

IV. On challenges and perspectives in avocado cultivation

40. What do you consider the main risks in avocado production?

- Overproduction in the region
- Competition with other areas of the country
- Competition with other countries
- Others

41. Problems with unfavorable weather conditions in the region

- Hail
- Frost
- Excess rain
- Lack of rain
- Increase in temperature
- Others

42. Problems with conditions associated with consumption

- Decrease in national consumption
- Market saturation
- Decrease in market prices
- Others

43. How many people work in the orchard?
44. How long did their work in a year?
45. How do you consider the access roads to your orchard?
46. In general terms, how would you consider the effect that avocado cultivation has had in economic terms in the region?
47. In general terms, how would you consider the effect that avocado cultivation has had in social terms in the region?
48. What would happen to you if the avocado markets declined or collapsed?
49. What do you think would happen to the region if the avocado markets declined or collapsed?
50. Do you observe impacts on water or soil contamination in your orchard in recent years?
51. Do you observe impacts on water or soil contamination in the region in recent years?

V. General information 52. Where are you from?

53. What is your production unit called?
54. Who do you consider to be the key people who started avocado cultivation in the region?
55. What is your principal occupation?

Date:
Place:
Name:
Age:

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
