# Peer review of "Socio-Environmental Impacts of the Avocado Boom in the Meseta Purépecha, Michoacán, Mexico"

_sustainability, doi:10.3390/su13137247_

Round 1

Reviewer 1 Report

Dear Authors,

The manuscript deals with the environmental and social impacts produced by the expansion of avocado cultivation in a region of central Mexico. The subject matter is of interest and appropriate for the journal. The expansion of the agricultural frontier in Latin America, closely associated with agribusiness, often has important consequences on the local communities and the landscape (land use and land cover) where they live. 

The manuscript presents a series of data from different sources, mostly reworked or combined for different purposes, to explain the process of avocado expansion and its consequences. The article is well written but is structured more like a report than a scientific article. In fact, no results of its own are presented, although a semi-structured interview-survey with key actors in avocado farming in the study area is mentioned (and presented as Appendix). There is even an overuse of footnotes, which should be avoided in a scientific article, or at least minimised. The results section does not include the discussion and then the discussion section does not have any reference to any of the references that have been discussed.

However, I believe that if the authors reorganise the structure of the article, it might even be more appropriate to present it as a review (expanding the references, too local) and, of course, including their own results from their interviews that have not been properly exploited in this version, the manuscript could perfectly well be resubmitted for review as the topic addressed is of great socio-ecological interest.

Kind regards,

Reviewer 2 Report

The paper deals with a very important and interesting topic, but it lacks scientific robustness. In particular, a specific paragraph should be devoted to the description of the methodology, that, at the moment, is too weak and descriptive. There are many structured and reliable methodologies to evaluate enviromental and social impacts, why you did not consider them? If the aim of the paper is to study just the perception of actors, than the questionnaire can be sufficient (but, it should be framed within the social research methodologies, recalling the most appropriate literature).
I hope to see very soon this paper improved and published because the topic treated is very important

Reviewer 3 Report

I consider it a good article that can give us an interesting insight into the socio-environmental impacts of avocado culture in Mexico.

Round 2

Reviewer 1 Report

Dear Authors,

I have carefully read the revised version of this manuscript and the authors' responses. I appreciate the effort taken in addressing my comments and the other reviewers' comments.  The authors have responded comprehensively and convincingly to all comments. The quality of the manuscript has improved greatly. 

From my perspective, the article is now ready to be published. However, while reading the revised manuscript I noted a few very minor issues so that a final read-through by the authors/editors might be useful. For example, 

Line 50 and Line 334: double full stop. 

Line 290: double semicolon and an open square bracket without text.

Lines 517-520 are in bold.

And probably several others...

Reviewer 2 Report

Dear Authors

The paper has been truly improved. However, I still have some observations to make:

- I suggest to read the following reference to improve the methodological section:

  • Bailey, Methods of social research
  • Bradburn, Sudman, Wansik. Asking Questions
  • Sociological research methods
  • Creswell and Creswell. Research Design. Qualitative, quantitative and mixed methods approaches

- the questionnaire attached to the paper cannot be qualified as “semistructured” because almost all questions are open

- the results section does not illustrate most of the results of the questionnaire. The section can be improved and enriched with graphs and figures. Information and data retrieved from primary and secondary sources should be illustrated separately, in my opinion.

- I suggest to better explain what is the snowball sampling (it fits very well in your case study, good choice), also adding bibliographic references
